# Omega-3 fatty acid normalizes postsynaptic density related miRNAs and proteins in hippocampus and prevents DEHP-induced impairment of learning and memory in mice

**Muyao Ding[1,2], Hongyu Ma[1], Hui Du[1], Yinglong Yang[1], Min Yu[1], Cong Zhang [3]***

**1** School of Public Health, Dalian Medical University, Dalian, P.R.China, **2** School of Public Health, Capital Medical University, Beijing, P.R.China, **3** Department of Food Nutrition and Safety, Dalian Medical University, Dalian, P.R.China

* congzhang1203@hotmail.com

## Abstracts

DEHP is the most widely used plasticizer in many products. However, growing evidence has indicated that DEHP may induce neurotoxicity. DEHP exposure affects mircoRNAs (miRNAs) expression in brain. A growing body of evidence suggests that nutrients and other bioactive food components prevent neurotoxicity through regulation of miRNA expression. Due to the increasing concern about the risks of DEHP to human health, we explored the neuroprotective effect of Omega-3 fatty acid (Omega-3FA) on subchronic DEHP-induced neurotoxicity in mice, and the potential involved miRNAs and their targets in the protective action of Omega-3FA against DEHP-induced neurotoxicity. Omega-3FA protected against the DEHP-induced impairment of learning and memory and alleviated the thinning of postsynaptic density (PSD) thickness in hippocampal synapses. We observed that there are fourteen up or down regulated miRNAs associated PSD in DEHP exposure which were normalized by Omega-3FA treatment. Protein targets in PSD of these differentially expressed miRNAs were predicted. Furthermore, the expression levels of protein mGluR5, Homer1, and NMDAR2B were carried out via Western blot, for further verifying PSD associated miRNAs' targets are involved in neuroprotection of Omega-3FA against DEHP. These findings suggested that Omega-3FA protected DEHP-induced impairment of learning and memory as well as synaptic structure alteration in the hippocampus by regulating the expression of PSD associated miRNAs and their targets. Thus, Omega-3FA should be included in diet to prevent or suppress neurotoxicity caused by continuous exposure to DEHP.

**Data availability statement:** All relevant data are within the paper and its Supporting Information files.

**Funding:** Undergraduate Innovation Project of Dalian Medical University (No. S202110161005). Research Project on Undergraduate Education Reform in Liaoning Province's General Higher Education (2018). The funders had no role in study design, data collection and analysis, decision to publish, or preparation of the manuscript.

**Competing interests:** The authors have declared that no competing interests exist.

## Introduction

Di-ethyl-hexyl-phthalate (DEHP) is widely used as plasticizer and softener in many consumer products, including building materials, food packaging, children's toys, medical devices, and cosmetics [1]. DEHP is not chemically bound to the end products, it may easily transfer into the environment and then enter the body through ingestion, inhalation and skin exposure [2,3]. Epidemiological studies have demonstrated that developmental DEHP exposure affected the neurodevelopment of newborns and decreased the intelligence of children [4,5]. Defects in learning tasks and changes in behavior were observed in animals exposed to DEHP [6]. These findings suggest that DEHP causes neurotoxic effects in the central nervous system, including learning and memory impairment.

Synaptic plasticity in the hippocampus is considered to be the basis of learning and memory. Changes in synaptic functional plasticity are tightly associated with alterations of synaptic structure [7, 8]. Postsynaptic density (PSD) is a protein-dense complex attached to the postsynaptic membranes, contains membrane receptors, scaffold proteins, kinases and many signaling molecules and is crucial in synaptic signal transduction [8,9]. The change in PSD thickness serve as an important indicator which reflects postsynaptic structural changes in hippocampus-dependent learning [10]. Several miRNAs are particularly abundant in the postsynaptic compartments, regulating synaptic plasticity and activity by modulating the translation of local proteins, including NMDAR, AMPAR, GTPase, cytoskeleton, and PSD scaffold proteins [11,12]. Moreover, PSD protein content is involved in a long-term increase in the synaptic strength [13]. Some studies have confirmed that miRNAs play a vital role in DEHP-induced neurotoxicity [14]. A Recent observation found that DEHP downregulated the expression of postsynaptic density protein 95 (PSD 95) and synapsin-1 [15], which play crucial roles in synaptic structure and function [16]. Therefore, focusing on miRNAs and their targets in PSD may provide insight into DEHP caused impairment of learning and memory.

As a major component in neuronal membrane, Omega-3 fatty acid (Omega-3FA) exhibits a wide range of regulatory functions [17], and play critical roles in maintaining brain structure. Omega-3FA deficiency has been related to hippocampal plasticity reduction and memory deficits in rodents, while dietary Omega-3FA supplementation may promote neuroplasticity and improve learning and memory abilities [18,19]. The increasing evidences have shown that Omega-3FA were effectively used in the treatment of neurodegenerative diseases or alleviation of neurotoxicity by exerting antioxidation, decreasing neuroinflammation, supporting synaptic density, and improving cognitive functions [20–23]. However, little is known about the function of Omega-3FA on the treatment of DEHP-related neurotoxicity. Since Omega-3FA supplementation has been explored against different environmental toxicant induced toxicity and has no known adverse effect [24,25], our study aimed to explore the protective effect of Omega-3FA against DEHP-induced learning and memory deficit, PSD associated miRNAs and their target proteins for new preventive or therapeutic opportunities.

## Materials and methods

### Animal treatment

Sixty KM mice (Male, 4-week-old) were obtained from the Experimental Animal Center of Dalian Medical University. Animal handling and procedures used in this study were approved by the Institutional Animal Care and Use Committee of Dalian Medical University (AEE21122). Five mice were housed in each cage with a 12h dark-light cycle, at 18°C-22°C and 50% humidity and were maintained on a standard diet with water available ad libitum. All procedures were carried out under strict accordance with National Institute of Health Guide for Care and Use of Laboratory Animals, and all efforts were made to minimize suffering.

The mice randomly divided into three groups, including control, DEHP (Sigma-Aldrich Corp, USA), and DEHP+Omega-3 FA (Aladdin, China) groups (n=20 in each group), they continuously received DEHP or DEHP+Omega-3 FA by gavage. Mice in the three groups were given 0.1mL of corn oil or corn oil with DEHP (5mg/kg) or DEHP (5mg/kg) + Omega-3 FA (150mg/kg, 18% EPA, 12% DHA) daily until the 12th week. The body weight of each mouse was weighed every week to adjust the dose of DEHP and Omega-3 FA. The concentration of DEHP exposure were set based on the Reference Dose ($R_fD$) set by the Environmental Protection Agency (20 times of the $R_fD$ for children) [26–28]. The dose of Omega-3 FA (150mg/kg) was founded on the studies carried out by Shaaban et al [29]. The human dose equivalent for the used animal dose of Omega-3 FA (150mg/kg) is approximately 18mg/kg [30], which meets the standard recommended daily doses for humans [31].

### MWM test

Morris water maze (MWM) performance is a standard technique in measure of hippocampally dependent spatial navigation and reference memory [32]. Briefly, the test was composed of the spatial acquisition phase and probe trial [33,34]. In spatial acquisition phase, a circular transparent escape platform located in the center of the N quadrant was submerged 1cm below the water surface. Animals were trained four sessions by starting at N, S, E, and W quadrants of the pool. The time spent to reach the platform (escape latency) within 60s was recorded as acquisition latency. The training of spatial acquisition phase last for four consecutive days.

On the fifth day, a spatial probe test was performed. The platform was taken away, each mouse was started at one point in the S quadrant and allowed to navigate freely in the pool for 60s. The crossings in the target quadrant (which had a hidden platform previously) were recorded by a smart video tracing system (NoldusEtho Vision system, version 5, Everett, WA, USA).

### Ultrastructure investigation

After 8 weeks treatment, there is not a single animal has died. The mice were anesthetized with sodium pentobarbital via i.p. injection. The brain tissue was removed, and hippocampus samples from each group were collected (−2mm at the anterior/posterior axis, ±1.8mm at the lateral/medial axis and −1.5mm at the dorsal/ventral axis), fixed with 4% glutaraldehyde, cooled and separated into 1mm³ pieces; which were later fixed, dehydrated, soaked, embedded and dual stained as previously described [35]. Finally, ultrathin hippocampus slices (500–700nm) were prepared and the ultrastructure of the synapses was observed via TEM (JEM-2000EX, Olympus, Tokyo, Japan), and the thickness and width of the PSD was recorded [36].

### miRNA sequencing and bioinformatic evaluation

Total RNA extracted from hippocampus tissue samples were used for library preparations and sequenced on an Illumina Hiseq 2500 platform (Illumina, San Diego, CA, USA). The biological significance of altered miRNA expression is closely related to their gene targets. Potential target genes of the differentially expressed miRNAs were predicted from data in the

databases: TargetScan, miRWalk, miRanda, miRNA.org and miRDB, and the final targets were integrated from at least two different programs. The GO functional annotation was carried out via DIANA-miRPath v3.0 [37–39].

### Real-time PCR analysis

miRNA was extracted from the mouse hippocampus tissue using RNAiso Plus according to the manufacturer's instructions (Takara, Japan). 1 μg of total RNA was reverse transcribed using a reverse transcription kit (Takara, Japan). Real-time PCR amplification was performed with a SYBR Green PCR kit (Takara, Japan) using the TP800 Real-Time PCR Detection System (Takara, Japan) as follow conditions: initial denaturation at 95 C for 3 min, followed by 40 cycles of 95°C for 12s, 60°C for 40s. The data were analyzed using the $2^{-\triangle\triangle CT}$ method.

### Western blot analysis

Western blot analyses were performed to detect the protein expression of mGluR5, Homer1, and NMDAR2B (based on results from miRNA screening). GADPH (housekeeping protein) was used as a control. The total protein was prepared as described previously [40], hippocampus samples were homogenized in ice cold RIPA Tissue Protein Extraction Reagent (Beyotime, China) supplemented with 1% proteinase inhibitor mix and incubated at 4°C for 1h. After incubation, debris was removed via centrifugation at 12,000 × g for 15 min at 4°C, and the lysates were stored at −80°C until being used. Protein concentration was determined using BCA assay. The samples employed for western blotting contained 50 μg of protein in each lane. Equal amounts of protein were separated via 10% SDS-PAGE gel and electrotransferred to Hybond-P polyvinylidene fluoride membranes (Millipore, France). The membranes were blocked with blocking buffer containing defatted milk powder for 1h and incubated overnight at 4°C with 1 μg/ml of primary antibodies against mGluR5 (1:1000, Abcam, USA), Homer1 (1:1000, Abcam, USA), DLG2 (1:1000, Abcam, USA) and NMDAR2B (1:1000, Abcam, USA). Then membrane was washed three times with Tris-buffered saline containing 0.05% Tween-20 (TBST) for 15 min and later incubated with horseradish peroxidase-conjugated goat anti-mouse IgG (1:7,000) (Sigma) at room temperature for 1 h. Enhanced ECL chemiluminescence was quantified densitometrically by UVP BioSpectrum Multispectral Imaging System (Ultra-Violet Products Ltd, Upland, CA).

### Statistical analysis

Data are presented as the mean±standard error of mean (SEM), and analyzed using SPSS 23.0 for Windows. One/two-way ANOVA followed by post hoc tests, were performed where appropriate, and in all instances a $p$ value of less than 0.05 was considered to be statistically significant.

## Results

### Omega-3FA improved DEHP-induced impairment of learning and memory

As the diagram shows the timeline of experimental protocols (Fig 1A), after 8-week exposure to DEHP, MWM test were performed, it has been employed in numerous studies demonstrating cognitive ability following changes of learning and memory induced by neurotoxins in the environment or nutrients in food in rodents [41]. The escape latency to identify the hidden platform and the times of crossing in the target quadrant are collected as the parameter of learning function and memory retention, respectively. As shown in Fig 1B, DEHP-induced learning and memory deficits were significantly improved by Omega-3FA. In the spatial acquisition phase, Omega-3FA-treated reduced escape latency compared with DEHP alone group (F (2,17) =19.79, $P < 0.001$, two-way ANOVA, Fig 1C). In the probe test, increased crossings over the platform in the target quadrant were showed in Omega-3FA-treated mice compared with DEHP alone group (Fig 1D). Though MWM test, we found that Omega-3FA could protect against DEHP-induced impairment of learning and memory.

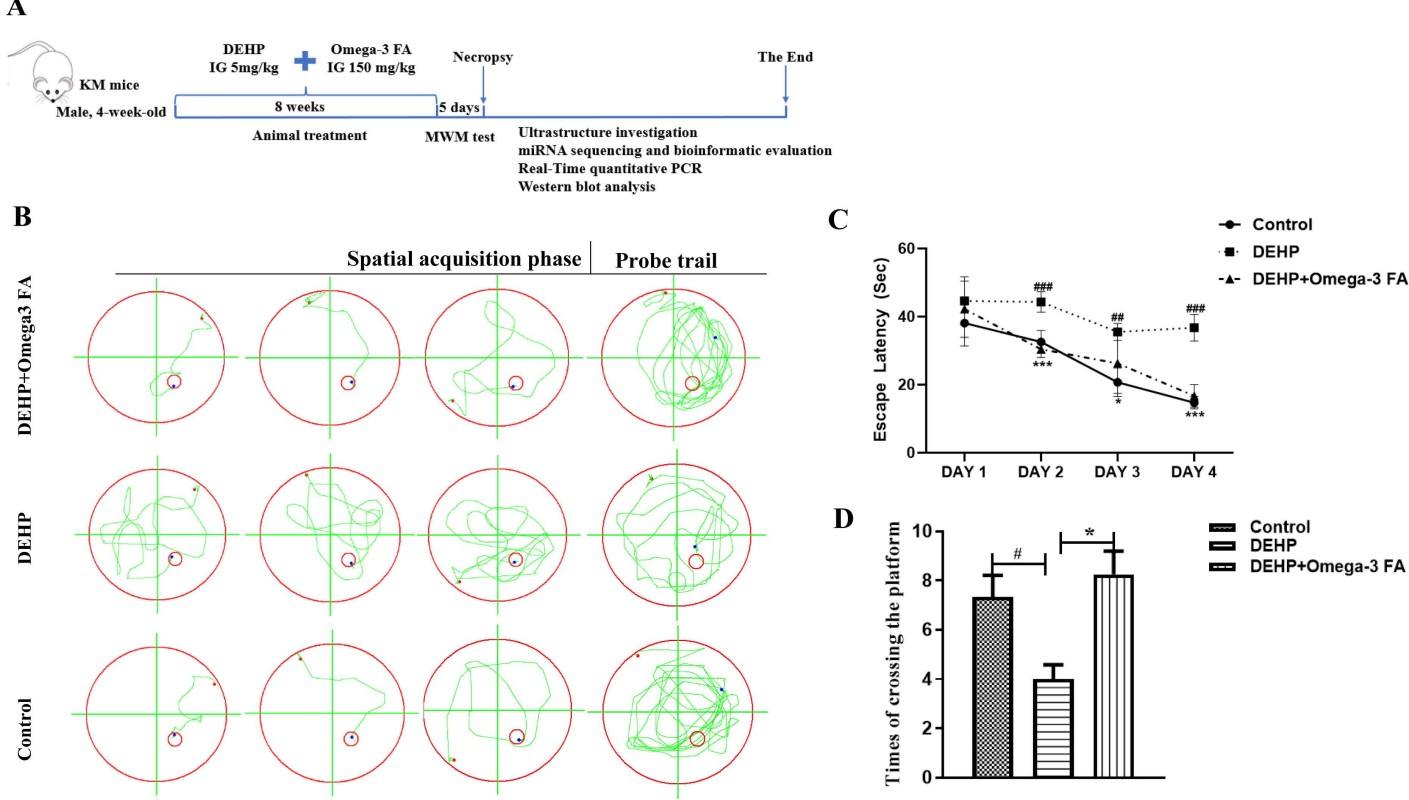

**Fig 1. Omega-3FA improved DEHP-induced impairment of learning and memory capacity in mice (n = 10-15).** The diagram shows the timeline of experimental protocols **(A)**. Representative swimming routes of each group during the spatial acquisition phase and probe trial in the Morris water maze (MWM) **(B)**. In spatial acquisition phase, the time spent to find the hidden platform of mice is shown **(C)**. In the probe trial, the total number of crossings the platform in the target quadrant of mice were shown **(D)**. ##$p < 0.01$, ###$p < 0.001$ compared to the control group; *$p < 0.05$, ***$p < 0.001$ compared to the DEHP group.

## Omega-3FA recovered the synaptic structure in hippocampus of DEHP-treated mice

To investigate whether Omega-3FA could protect against DEHP induced synaptic structural changes, we further examined the structural parameters of the hippocampus synaptic interface.

As shown in Fig 2A, the ultra-structure of hippocampus synapse was obviously recovered in DEHP+Omega-3FA group compared with DEHP alone group. Quantitatively, compared with the control group, decreased PSD thickness and the increased synaptic cleft were detected in the hippocampus synapse of the DEHP group. Treatment with Omega-3FA increased the PSD thickness and decreased the synaptic cleft width significantly compared with the DEHP group (Fig 2B).

## Omega-3FA affects postsynaptic density related miRNAs in alleviating DEHP-induced impairment of learning and memory

Differentially expressed miRNAs were shown in Fig 3A, there were 62 differently expressed miRNAs identified between the DEHP group and control group, as well as there were 89 differently expressed miRNAs between the DEHP+Omega-3FA group and the DEHP group. We found 25 common miRNAs on both lists of differentially expressed miRNAs in DEHP vs CON and DEHP+Omega-3FA vs DEHP groups (Fig 3B). Excluding those with changes in the same direction, we identified twenty miRNAs, upregulated miR-3963, miR-344-3p, miR-335-3p,

**A**

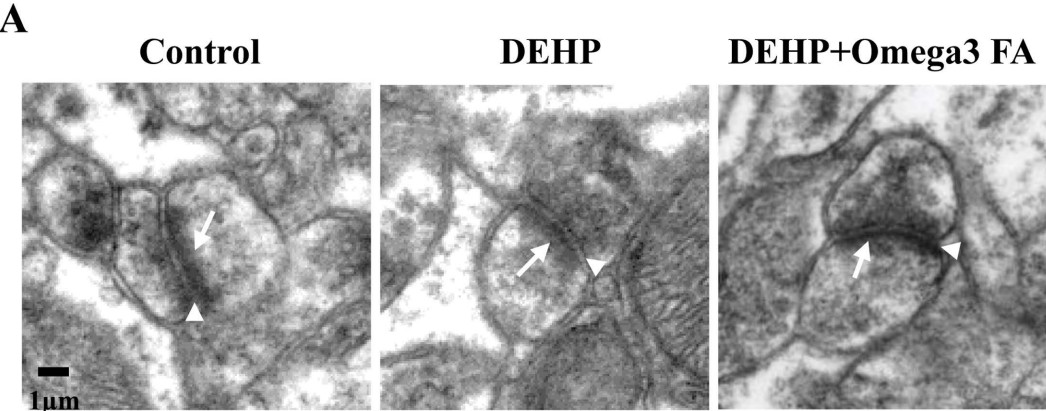

**B**

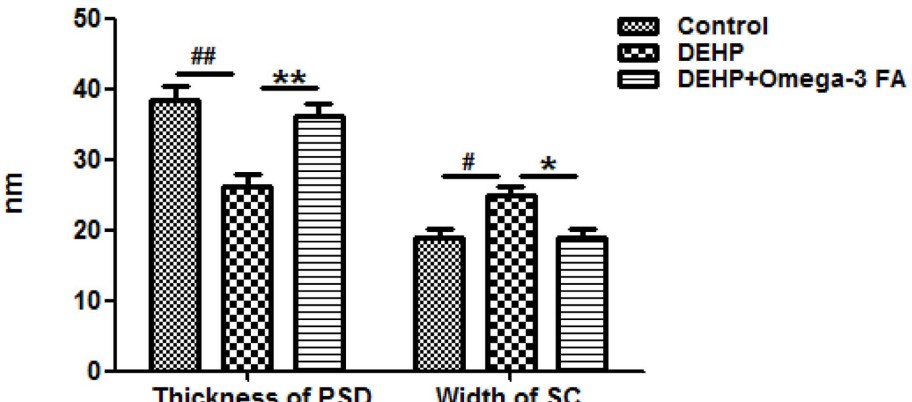

**Fig 2. Effects of Omega-3FA on synaptic ultrastructure in hippocampus of mice exposed to DEHP (n = 5).** Comparison of synaptic structure in the mice hippocampus among groups **(A)**. PSD, postsynaptic density (arrows); SC, synaptic cleft (arrowheads). Quantitatively changes in the PSD thickness and SC width of synaptic in the mice hippocampus among groups **(B)**. #$p < 0.05$, ##$p < 0.01$ compared to the control group; *$p < 0.05$, **$p < 0.01$ compared to the DEHP group.

let-7a-5p, let-7g-5p, miR-145a-5p, miR-122-3p, miR-652-3p, miR-381-3p and miR-26b-5p in DEHP exposure which were downregulated by Omega-3 FA treatment, and downregulated miR-129b-5p, miR-378b, miR-411-3p, miR-338-5p, miR-301a-3p, miR-1195, miR-6238, miR-1970, miR-1969, and miR-690 which were upregulated by Omega-3 FA treatment (Fig 3C).

After bioinformation analysis of these twenty miRNAs, we focused on the changed miRNAs associated with GO term (GOTERM CC FAT) of "Postsynaptic density" (Table 1). In Table 2, Fourteen differentially expressed miRNAs linked "Postsynaptic density" in DEHP vs CON and DEHP+Omega-3FA vs DEHP groups were listed out, including miR-344-3p, miR-335-3p, let-7a-5p, let-7g-5p, miR-145a-5p, miR-381-3p, miR-26b-5p, miR-129b-5p, miR-378b, miR-338-5p, miR-6238, miR-1970, miR-1969 and miR-690. We selected miRNAs (miR-335-3p, miR-26b-5p, miR-338-5p and miR-690) that regulate three or more PSD related proteins for validation using Real-time PCR, the expression levels of these miRNAs displayed a similar regulation trend to the gene sequencing results (Fig 4).

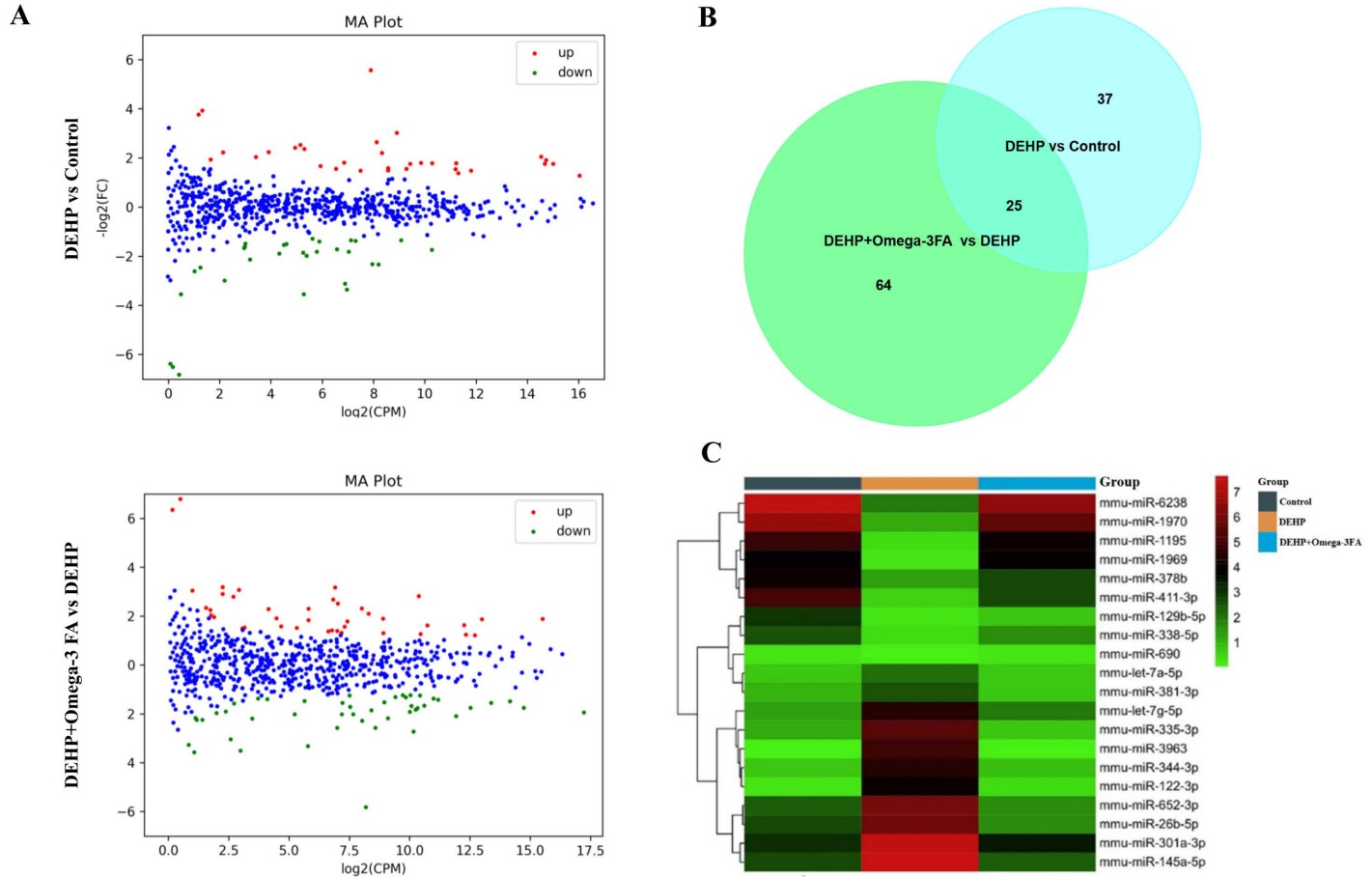

**Fig 3. Detail information of differentially expressed miRNAs (n = 3). (A)** Scatter plot of differentially expressed miRNAs in hippocampus. Red and green dots indicate up and downregulation of miRNA, respectively, relative to the control (*P* < 0.05). **(B)** Venn diagram showing the number of significantly modulated miRNAs in DEHP vs control and DEHP+Omega-3 FA vs DEHP. **(C)** Heat map representation of significantly differentially expressed miRNAs (group averages) in control, DEHP and DEHP+Omega-3 FA-treated mice.

**Table 1. The GO terms significantly changed in cellular component (GOTERM CC FAT) of differentially expressed miRNAs in DEHP vs CON and DEHP+Omega3FA vs DEHP groups.**

| Term | *p*-Value | Ontology |
|---|---|---|
| Synapse | 5.0E-9 | Cellular component |
| Dendrite | 1.3E-6 | Cellular component |
| Dendritic spine | 2.2E-5 | Cellular component |
| Postsynaptic density | 1.2E-4 | Cellular component |
| Neuronal cell body | 1.8E-4 | Cellular component |
| Presynaptic membrane | 3.0E-3 | Cellular component |
| Cytoskeleton | 9.4E-3 | Cellular component |
| Postsynaptic membrane | 1.5E-2 | Cellular component |

**Table 2.** Postsynaptic density related differentially expressed miRNAs in DEHP vs CON and DEHP+Omega-3 FA vs DEHP groups.

| Postsynaptic density related differentially expressed miRNAs | DEHP vs CON *(Upregulated)* | miR-344-3p miR-335-3p let-7a-5p let-7g-5p miR-145a-5p miR-381-3p miR-26b-5p |
| --- | --- | --- |
| | DEHP+Omega-3 FA vs DEHP *(Downregulated)* | |
| | DEHP vs CON *(Downregulated)* | miR-129b-5p miR-378b miR-338-5p miR-6238 miR-1970 miR-1969 miR-690 |
| | DEHP+Omega-3 FA vs DEHP *(Upregulated)* | |

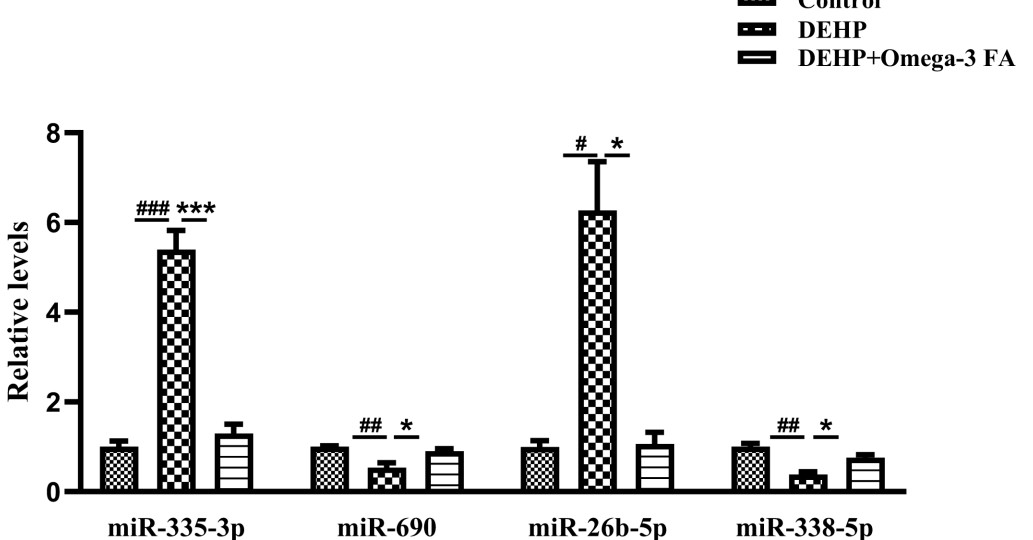

**Fig 4. The differential relative expression of PSD related miRNAs in hippocampus were validated by Real-Time PCR (n = 3).** #$p < 0.05$, ##$p < 0.01$, ###$p < 0.001$ compared to the control group; *$p < 0.05$, ***$p < 0.001$ compared to the DEHP group.

## Post synaptic density related miRNAs' target genes analysis and conformation

The potential mRNA targets of these differentially expressed miRNAs associated with "Postsynaptic density" were further collected using the comparative platforms. Targets calculation was performed by at least 2 different algorithms in case of minimizing the number of putative and maybe false positive targets. Target mRNAs of the PSD related miRNAs were predicted and shown in Fig 5.

The protein expression levels of the target gene GRM5, HOMER1, DLG2 and GRIN2B were further investigated, due to the targeted regulation of more than 2 differentially expressed miRNAs. The expression levels of the mGluR5 (GRM5), Homer1 (HOMER1), DLG2 and NMDAR2B (GRIN2B) proteins in hippocampus are shown in Fig 6. DEHP exposure reduced the expression levels of mGluR5, Homer1, DLG2 and NMDAR2B protein in hippocampus of mice, which was significantly upregulated by Omega-3FA.

## Discussion

In the present study, we show for the first time that Omega-3FA provides protect effects on DEHP-induced neurotoxicity. Our research has revealed that Omega-3FA supplement could modulate PSD related miRNAs' expression and improved synaptic structure as well as learning and memory ability, which were weakened by DEHP.

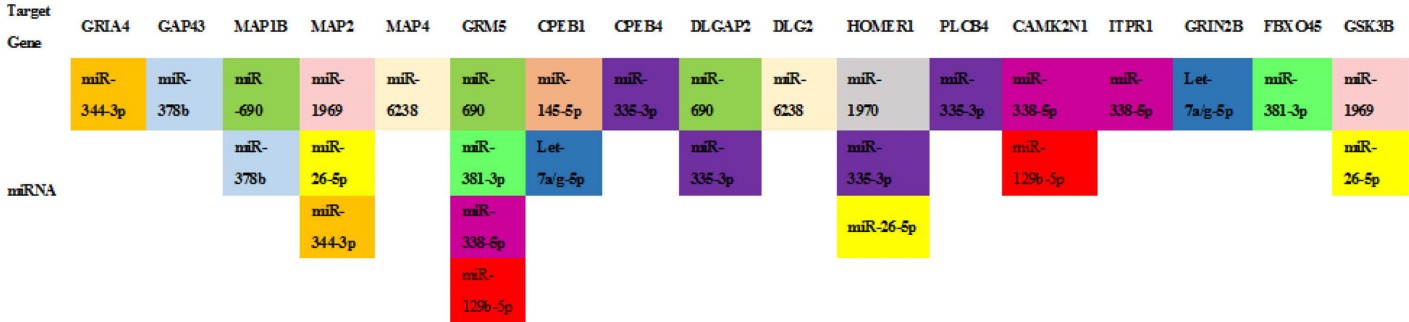

**Fig 5. Detail information of** postsynaptic density **(PSD) regulatory miRNAs' target genes in PSD (The same miRNAs were filled with one color).**

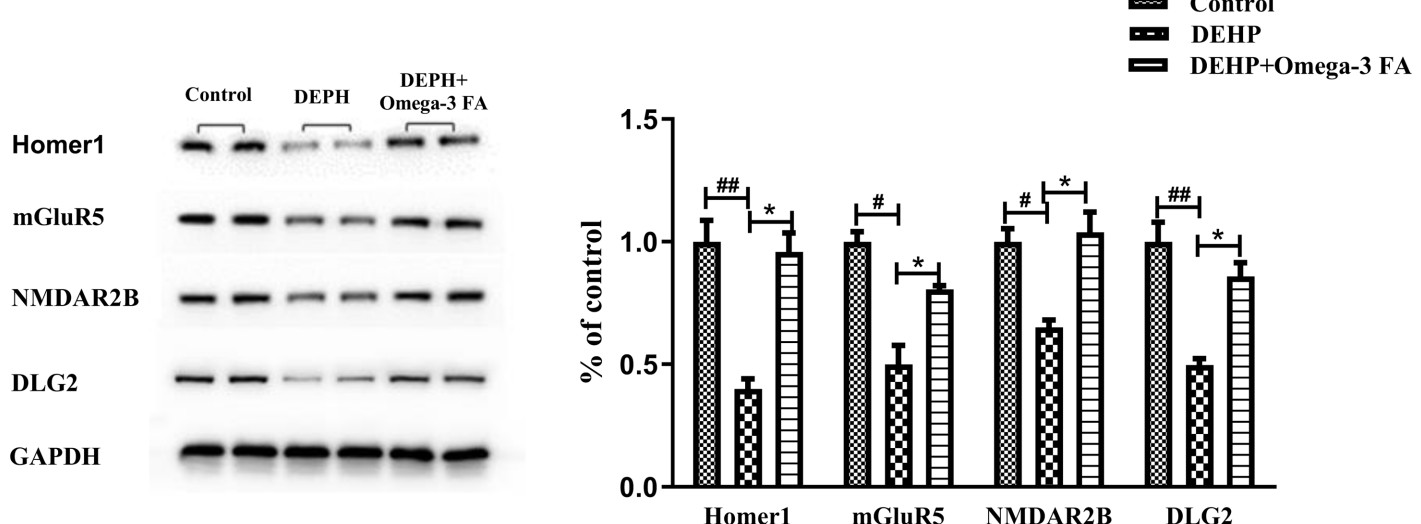

**Fig 6. Western blot analysis of the protein levels of Homer1, mGluR5, NMDAR2B and DLG2 in hippocampus (n = 5).** [#]$p < 0.05$, [##]$p < 0.01$ vs the control group; [*]$p < 0.05$ vs the DEHP group.

DEHP is a lipophilic compound and well absorbed after oral exposure. In human body, the absorption of DEHP could be 25% [42]. DEHP may induces neuronal degeneration and neurobehavioral abnormalities through various mechanisms or pathways, including the alteration of trace element and mineral levels, reduction of the testosterone levels, induction of endoplasmic reticulum (ER) stress, promotion of inflammatory parameters, or even the cortex may be the site major affected by hydroxymethylation [27,43–45]. A recent study investigated that high dose DEHP affected the integrity the blood-brain barrier (BBB), which could lead to neurotoxic effects [46]. Omega-3FA is associated with BBB integrity in human [47]. In animal models, Omega-3FA mitigate BBB disruption after brain injury [48]. Furthermore, DEHP altered the lipid metabolome in the fetal brain, down-regulated the lipid profiling especially the composition of DHA, an important constituent of Omega-3FA [44,49]. Dong et al. recently reported that maternal DEHP exposure affected the synaptic ultra-structure of hippocampus in male offspring [8]. In embryonic neuronal cultures, DHA-deficient reduced long-term poten-tiation [50]. Oral DHA supplementation increased the number of dendritic spines in adult gerbil hippocampus [51]. The thickness of PSD is a crucial indicator of synapses morphology, as the efficiency of synaptic transmission decreases when the PSD becomes thinner [52]. In this study, we found that Omega-3FA administration increased thickness of PSD, which was associated with alleviated cognitive impairment in mice of DEHP exposure. Therefore, it is suggesting that variations

in the components and structure of the PSD underpin abnormal synaptic plasticity and ultimately lead to neurological disorders after DEHP exposure.

miRNAs have been identified as potential diagnostic and therapeutic targets in many diseases [53], and recently have emerged as key regulators of neuronal development and synaptic plasticity [54]. One previous study has verified that the levels of various miRNAs are significantly changed in brain tissues after DEHP exposure [14]. It was reported that Omega-3FA affected miRNA expression in hippocampus of animal models [55]. Therefore, we analyzed the effect of Omega-3FA on miRNAs expression after DEHP exposure. Our bioinformatics analysis on differential expression miRNAs showed that "Post synaptic density" was an important cellular component of Omega-3FA protecting against DEHP neurotoxicity. There were fourteen differential expression miRNAs presenting in the "Post synaptic density" term, including: miR-344-3p, miR-335-3p, let-7a-5p, let-7g-5p, miR-145a-5p, miR-381-3p, miR-26b-5p, miR-129b-5p, miR-378b, miR-338-5p, miR-6238, miR-1970, miR-1969 and miR-690. Variation in PSD associated miRNAs expression affects level of synaptic proteins, and consequently modulating synaptic plasticity through the stability and translation of dendritically localized transcripts [56]. As a multimodal hub, PSD is located on the postsynaptic membranes at the synaptic junction, composed of hundreds of different proteins such as scaffold proteins, glutamate receptors, calmodulin binding protein, ion channels, and signaling molecules [57]. These protein components and their interactions may elucidate the mechanism of long-term changes in synaptic plasticity, which underlie learning and memory [58].

In our study, DEHP exposure significantly reduced protein levels of Homer1, mGluR5, NMDAR2B and DLG2 in the hippocampus of mice, which were increased under administration of Omega-3FA, which may associate with improved synaptic structure and behavior. Homer is one of the most widely studied PSD scaffold proteins that closely involved in mechanisms underlying hippocampus-dependent memory processes [59]. Homer proteins are including the transcripts of three mammalian genes, namely Homer 1, Homer 2 and Homer 3, each with many splice variants including the short isoforms Homer1a and Ania3 and long isoforms Homer1b/c and Homer2 and Homer 3 [60]. Homer 1-knockout mice exhibit learning deficits [61]. Homer1a, which is produced by an immediate early gene, modulates ligand-independent type I metabotropic glutamate receptor (mGlu5R) signaling [62]. Homer1b anchors mGlu5R to NMDARs and several signaling molecules, such as transient receptor potential ion channels and inositol-1,4,5-trisphosphate receptors [63,64]. Spatial memory of normal rats can be augmented when Homer 1c is overexpressed in hippocampus [65]. Therefore, Homer 1 protein acts by linking group I mGluRs to NMDARs, as well as by bridging mGluRs with their intracellular downstream effectors [66]. Recent studies have focused on Homer1 exposure to environmental toxicants in rodents [59]. Our result suggested that GRM5/Homer1/NMDAR2 can be affected by DEHP in neuronal tissues. It was found that NMDAR-dependent LTP and mGlu5R-mediated LTD were lacking in adult Omega-3FA-deficient mice [67]. In perinatal mice, dietary Omega-3FA depletion decreased the Homer1 expression, which could be normalized when the Omega-3FA supplemented in time [68]. These results suggest that Omega-3FA protected synaptic plasticity underlying learning and memory ability, related to normalize the expression of GRM5/Homer1/NMDAR2.

In this study, we noticed that Homer1 gene could be targeted by miR-335-3p, miR-1970 and miR-26-5p; mGluR5 gene could be targeted by miR-690, miR-381-3p, miR-338-5p and miR-129-5p; as well as NMDAR2B gene could be targeted by Let-7a/g-5p. It has been confirmed that miR-335 is involved in synaptic plasticity [69]. Shi et al. found that miR-129-5p directly targeting GRM5 [70]. Our results imply GRM5/Homer1/NMDAR2 related miRNAs which associate with synaptic plasticity maybe an important clue for exploring the protective molecular mechanism of Omega-3FA in DEHP induced neurotoxicity.

In addition to the above, our bioinformatics analysis provides more information on possible research clues. NMDAR2B subunit has PDZ-binding domains on their extreme C-termini that are known to interact with the all four members of PSD-95 family and other PDZ proteins [71]. Previous studies have suggested that DEHP alters expression of PSD-95 [15]. PSD95 is interact with DLG2 proteins and having in the role of glutamate levels, the DLG2 expression level were increased under administration of Omega-3FA. Together with our results, it suggested that DEHP affected PSD-95 might

be associated with NMDAR2B. MAPs are a group of proteins with either enzymatic or structural activity, which can interact with tubulin polymers. GSK3β is associated with expression of microtubule-protein such as MAP1B, MAP2, APC, CRMP2, and tau [72]. Sun et al suggested that perinatal DEHP exposure affect the GSK-3β expression and increased level of phospho-Tau in hippocampus [73]. In our present study, GSK3β can be regulated by miR-1969 and miR-26-5p; MAP1B, MAP2 and MAP4 which were targeted by miR-1969, miR-26-5p, miR-344-3p, miR-690, miR-378b and miR-6238. Functional networks of miRNAs and their target genes in synapses is complex, as each miRNA may target many mRNAs and each gene could be regulated by multiple miRNAs [74]. The molecular processes in PSD that underlie synaptic plasticity is believed to control the intracellular cross-linking amongst the transduction pathways.

Taken together, these observations highlight that DEHP exposure could disturb miRNAs expression and affect the levels of PSD proteins, leading to changes in the morphology of hippocampus postsynaptic density, which may be involved in the mechanisms of DEHP induced learning and memory defects. The protective efficacy of Omega-3FA on neurotoxicity caused by DEHP via modulating expression of PSD related miRNAs and proteins suggests possible targets of treating the neurotoxicity of DEHP as well as Omega-3FA might be a promising candidate for future study.

## Supporting information

**S1. Raw_images of Western blot.**
(PDF)

**S2. Differentially expressed genes (DEHP-Brain vs CON-Brain).**
(XLSX)

**S3. Differentially expressed genes (DEHP+Omega3 FA-Brain vs DEHP-Brain).**
(XLSX)

## Author contributions

**Conceptualization:** Cong Zhang.

**Data curation:** Yinglong Yang, Min Yu.

**Formal analysis:** Yinglong Yang, Min Yu.

**Funding acquisition:** Muyao Ding.

**Investigation:** Muyao Ding, Hongyu Ma, Hui Du.

**Project administration:** Cong Zhang.

**Writing – original draft:** Muyao Ding, Cong Zhang.

**Writing – review & editing:** Cong Zhang.

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
