## [Decision Letter · Decision Letter 0]

Dear Dr. Zhang,

Thank you for submitting your manuscript to PLOS ONE. After careful consideration, we feel that it has merit but does not fully meet PLOS ONE’s publication criteria as it currently stands. Therefore, we invite you to submit a revised version of the manuscript that addresses the points raised during the review process.

We look forward to receiving your revised manuscript.

Kind regards,

Pankaj Kumar Arora, Ph.D

Academic Editor

PLOS ONE

Journal Requirements:

-2,5-hexanedione-induced deregulation of axon-related microRNA expression in rat nerve tissues (https://doi.org/10.1016/j.toxlet.2019.11.019)

(among others)

In your revision ensure you cite all your sources (including your own works), and quote or rephrase any duplicated text outside the methods section. Further consideration is dependent on these concerns being addressed.

Reviewers' comments:

Reviewer's Responses to Questions

**Comments to the Author**

1. Is the manuscript technically sound, and do the data support the conclusions?

Reviewer #1: Partly

Reviewer #2: Yes

Reviewer #3: Yes

Reviewer #4: Partly

2. Has the statistical analysis been performed appropriately and rigorously?

Reviewer #1: Yes

Reviewer #2: N/A

Reviewer #3: Yes

Reviewer #4: Yes

3. Have the authors made all data underlying the findings in their manuscript fully available?

Reviewer #1: No

Reviewer #2: Yes

Reviewer #3: Yes

Reviewer #4: Yes

4. Is the manuscript presented in an intelligible fashion and written in standard English?

Reviewer #1: Yes

Reviewer #2: Yes

Reviewer #3: Yes

Reviewer #4: Yes

Reviewer #1: This work compared the learning and memory abilities, PSD microstructures, miRNA expression profiles and PSD-related proteins of the mice treated with DEHP or DEHP+Omega-3FA. They concluded that Omega-3FA protected DEHP-induced impairment of learning and memory as well as synaptic structure alteration in the hippocampus by regulating the expression of PSD associated miRNAs and their targets. This manuscript is well written. The findings regarding the beneficial effects of Omega-3FA on the DEHP-induced impairments of memory and PSD structure are novel and interesting. However, the contributions of the molecular changes to behavioral and cellular changes are poor, which can not support their conclusion described as "... by regulating the expression of PSD associated miRNAs and their targets". Some major issues are as follow:

1) Changes of some miRNAs and proteins were observed in the hippocampi of the mice with DEHP or DEHP+Omega-3FA treatment. There lacks experimental evidence to support the regulation of PSD proteins by those changed miRNAs. In addition, a causal relationship between differentially expressed (DE)-miRNAs and PSD structure alteration is not determined.

2) I suggest that the levels of some important DE-miRNAs between these groups should be compared by qPCR experiments, and some of them may also be confirmed to localized at synapses by fluorescence in situ hybridization (FISH) method.

3) Only DE-miRNAs were shown in the present version, additional data regarding miRNA seqencing should be prepared as supplementary files.

4) As descibed in method section, they made GO function and KEGG pathway analyses using the identified DE-miRNAs. Some enriched GO funtions (Cellular Component) were listed in Table 1. Where is the data from the enriched KEGG pathways?

Reviewer #2: In the present study, authors have investigated the potential effect of omega-3 fatty acids on DEHP-induced neurotoxicity in mice using molecular and behavioral methods. Overall, the subject is timely and interesting, methodology is solid and results are clear to the readers, however, I propose the following concerns to be addressed prior to publication of this research:

Major concerns:

1. Ethical considerations/protocols must be clarified in the manuscript (ethics code/number etc.)

2. Please describe the combination of O3 fatty acids in details (type and proportion of fatty acids used in the study). By the way, dose of O3 (150 mg/kg) should be justified reasonably in discussion.

3. I strongly suggest to present the results of MWM during time (trial period in Days) and then re-analyze the results by two-way ANOVA. This would enable the reader to more deeply understand how memory function is affected by learning and treatments.

4. I expect the authors to provide a detailed graphical timeline demonstrating the main experimental protocols in a sequential manner.

5. In Introduction and/or Discussion, authors have ignored recent literature on the effect of omega-3 fatty acids on memory function (please see https://doi.org/10.1016/j.lfs.2023.122100, and https://doi.org/10.1002/jdn.10336).

Minor concerns:

1. Please use “Control” instead “Con” in figures.

2. The entire manuscript required revision for English (mainly consistency and clarity of Discussion).

Reviewer #3: Introduction: This part is far too vague and contains too many generalities. There is a need to better explain.

Please provide the volume of DEHP and Omega-3 FA injectate administered for gavage.

Did the control group receive normal saline gavage?

Western blotting: Write homogenization process in detail mentioning the amount of tissue homogenized, volume and pH of buffer used, temperature for homogenization and include the make and model of the homogenizer. How much protein was loaded on the gel? Mention incubation time with primary and secondary antibodies. It is advised to specify each antibody used (primary and secondary), its concentration and the provider.

Reviewer #4: The author was well written the manuscript. Further, I picked up on a few additional points for consideration and correction:

1. Did you use the behavioral assessment rats for further neurochemical studies or not? If you did not use these rats, then explain what happened with behavioral assessment rats.

2. Animal ethical committee approval number should be mention in the methodology section.

3. Justification of DEHP doses is lacking in the manuscript, mention in the methodology section.

4. PSD95 is interact with DLG2 proteins and having in the role of glutamate levels, author need to check the DLG2 expression levels as well as the glutamate levels.

5. How much amount of DEHP have reached to brain, it has to be discussed about the levels of DEHP is responsible for these alterations in the brain.

**Do you want your identity to be public for this peer review?** For information about this choice, including consent withdrawal, please see our Privacy Policy

Reviewer #1: No

Reviewer #2: No

Reviewer #3: No

Reviewer #4: No

---

## [Author Response · Author response to Decision Letter 1]

26 Apr 2025

We thank you very much for giving us an opportunity to revise our manuscript. We are grateful to the editors and reviewers for the hard work in the paper titled “Omega-3 fatty acid normalizes postsynaptic density related miRNAs and proteins in hippocampus and prevents DEHP-induced impairment of learning and memory in mice”. We also appreciated for the advice given by the reviews. In the revised paper, we tried our best to answer the questions of the academic editor and reviewers, and provided a point-by-point response with amendments highlighted in yellow. Our blot image data are in Supporting Information (S1-raw-images).

We are very sorry for the carelessness in uploading our manuscript last time. Our fund should be following:

Undergraduate Innovation Project of Dalian Medical University (No. S202110161005).

Research Project on Undergraduate Education Reform in Liaoning Province's General Higher Education (2018).

---

## [Decision Letter · Decision Letter 1]

Omega-3 fatty acid normalizes postsynaptic density related miRNAs and proteins in hippocampus and prevents DEHP-induced impairment of learning and memory in mice

PONE-D-24-38755R1

Dear Dr. Zhang,

We’re pleased to inform you that your manuscript has been judged scientifically suitable for publication and will be formally accepted for publication once it meets all outstanding technical requirements.

Kind regards,

Pankaj Kumar Arora, Ph.D

Academic Editor

PLOS ONE

Additional Editor Comments (optional):

Reviewers' comments:

Reviewer's Responses to Questions

**Comments to the Author**

Reviewer #1: All comments have been addressed

Reviewer #2: All comments have been addressed

2. Is the manuscript technically sound, and do the data support the conclusions?

Reviewer #1: Yes

Reviewer #2: Yes

3. Has the statistical analysis been performed appropriately and rigorously?

Reviewer #1: Yes

Reviewer #2: Yes

4. Have the authors made all data underlying the findings in their manuscript fully available?

Reviewer #1: Yes

Reviewer #2: Yes

5. Is the manuscript presented in an intelligible fashion and written in standard English?

Reviewer #1: (No Response)

Reviewer #2: Yes

Reviewer #1: I am glad that the authors have done their best to address most of my concerns. The revised version could be accepted for publication.

Reviewer #2: Authors have satisfactorily improved their manuscript. I find this work now worthy of publication in PLOS One.

**Do you want your identity to be public for this peer review?** For information about this choice, including consent withdrawal, please see our Privacy Policy

Reviewer #1: No

Reviewer #2: No

---

## [Editor Report · Acceptance letter]

PONE-D-24-38755R1

PLOS ONE

Dear Dr. Zhang,

I'm pleased to inform you that your manuscript has been deemed suitable for publication in PLOS ONE. Congratulations! Your manuscript is now being handed over to our production team.

Kind regards,

on behalf of

Dr. Pankaj Kumar Arora

Academic Editor

PLOS ONE